# On the Consensus Performance of Multi-Layered MASs with Various Graph Parameters—From the Perspective of Cardinalities of Vertex Sets

**DOI:** 10.3390/e25010040

**Published:** 2022-12-26

**Authors:** Da Huang, Zhiyong Yu

**Affiliations:** 1Department of Mathematics and Physics, Xinjiang Institute of Engineering, Urumqi 830023, China; 2College of Mathematics and System Science, Xinjiang University, Urumqi 830017, China

**Keywords:** multi-agent systems (MASs), consensus, multi-partite graph, multilayered networks, Laplacian spectrum

## Abstract

This work studies the first-order coherence of noisy multi-agent networks with multi-layered structures. The coherence, which is a sort of performance index of networks, can be seen as a sort of measurement for a system’s robustness. Graph operations are applied to design the novel multi-layered networks, and a graph spectrum approach, along with analysis methods, is applied to derive the mathematical expression of the coherence, and the corresponding asymptotic results on the performance index have been obtained. In addition, the coherence of these non-isomorphic multi-layered networks with three different graph parameters are compared and analyzed. We find that, when the cardinalities of the vertex sets of corresponding counterpart layers are the same, the multi-layered topology class with a balanced, complete, multi-partite structure has the best robustness of all the considered networks, if the sufficient conditions for the node-related parameters hold. Finally, simulations are given to verify the asymptotic results.

## 1. Introduction

The consensus problem is a significant interdisciplinary field of MASs. It requires that all nodes in the networked system interact with each other based on the communication links and control protocols, so that they can achieve the desired physical states.

Researchers have done lots of significant work on consensus from various perspectives, such as the system order [1,2,3,4,5,6,7,8,9,10,11,12,13,14,15,16,17,18,19,20,21,22] (first/higher order or fraction order [7]), patterns of communication (continuous or discontinuous [4,8]), control methods (adaptive control [6,7,8], intermittent control [9]), and convergence time (finite time or fixed time [5,10]).

In the problem of complex networks, the interactive relationship in the networked system can be always interpreted by the topological structures [22,23,24], and some performance index of the system, such as consensus speed [1,11] or network coherence [13,14,15,16,17,18,19,20,21], can be characterized by the second smallest Laplacian eigenvalue or Laplacian spectrum, respectively. Similarly, synchronization problem can also be studied from the aspect of system structures [23,24,25,26,27,28,29,30].

In the past two decades, there has been lots of significant enlightening research on the Laplacian eigenvalues [1,12,13,14,15,16,17,18,19,20,21,22,31,32,33] of the networked systems. Reference [12] has studied the mathematical relations between the Laplacian eigenvalues and robustness, and derived the $H_{2}$ norms of some classic network structures. The robustness of the noisy systems can be defined by network coherence, and the notion was proposed in [13,14]. These articles discovered the fact that the coherence can be characterized by the nonzero Laplacian eigenvalues. Reference [16] studies the robustness of a noisy, scale-free, small-world Koch network, and shows that coherence relies on the shortest-path distance between the lead vertex and the largest-degree vertices. In [19], an analytical expression for the leader–follower coherence is determined that depends on the numbers of leaders and network parameters.

Recently, multilayered structures became a hot research topic of complex networks [22,25,26,27,31,32,33] because of their potential applications. It is natural and meaningful to extend the Laplacian-spectrum approach to the consensus problems with multi-layered structures. The multi-partite graph is also a significant structure in many fields, such as heterogeneous networks [28], the field of graph searching [29], and electric networks [30].

Inspired by the above facts, this research considers multi-layered MASs with specific topological structures that are constructed by graph operations. In addition, another sort of multi-layered systems with classic graph as subgraphs are studied for contrast, since they all have a fan shaped structure as their layered graph from the vertical sight. Specifically, the novelties are listed as follows:I.A sort of novel multi-layered MAS with a balanced, complete, multi-partite graph has been constructed by graph operations. Different from other good research on multilayered coordination systems, the side structure in this article from the vertical view has the fan-shaped structure.II.Analysis methods with various parameters are applied for deriving the coherence, and the related asymptotic properties have been acquired.III.We found that when the vertex sets of the corresponding counterpart layers have the same cardinality, the multi-layered graph class with a complete multi-partite structure has the best robustness of all the considered layered systems if the sufficient conditions for the parameters hold.

The aims for this paper were to study the robustness related index of the multi-layer MAS with noise and determine the mathematical expression for the coherence, and to further calculate the corresponding limitation-related results.

In Section 2, the needed notation of graph theory is explained, and the mathematical relation for the Laplacian eigenvalues and the coherence is described. Section 3 describes the topological structures of the multi-layered networks, and the derivations for the coherence are given. The simulated results are presented in Section 4.

## 2. Preliminaries

### 2.1. Graph Theory and Notations

Denote the complete graph with *n* vertices by $K_{n}$, and the star graph with k−1 leaf vertices is represented as $S_{k}$. $E_{k}$ denotes the empty graph with *k* vertices. Denote the fan graph with *q* vertices by $F_{q}$. Let *G* be a graph with vertex set $V=\{v_{1},v_{2},\ldots ,v_{N}\}$, and its edge set is defined as $E=\{\left(v_{i},v_{j}\right)|i,j=1,2,\ldots ,N;i\neq j\}$ The adjacency matrix is defined as $A\left(G\right)={\left[a_{ij}\right]}_{N}$, where $a_{ij}$ is the weight of $\left(v_{i},v_{j}\right)$. This paper considers the undirected graph, in which the edge weight satisfies: $a_{ij}=a_{ji}$, and it is supposed that $a_{ij}=1$, if $\left(v_{i},v_{j}\right)\in E;a_{ij}=0,$ if $(v_{i},v_{j})\notin E$. The Laplacian matrix of *G* is denoted as L(G)=D(G)−A(G), where D(G) is defined by $D\left(G\right)=diag\left(d_{1},d_{2},\ldots ,d_{N}\right)$ with $d_{i}=\sum_{j\neq i}a_{ij}$. The Laplacian spectrum of *G* is denoted by $SL\left(G\right)=\left(\begin{array}{cccc} \lambda_{1}\left(G\right) & \lambda_{2}\left(G\right) & ... & \lambda_{p}\left(G\right)\\ l_{1} & l_{2} & ... & l_{p} \end{array}\right)$, where $\lambda_{1}\left(G\right)<\lambda_{2}\left(G\right)<...<\lambda_{p}\left(G\right)$ are the Laplacian eigenvalues, and $l_{1},l_{2},\ldots ,l_{p}$ are the multiplicities of the eigenvalues [34].

In addition, the following definitions and lemmas are needed for deriving the results.

**Definition** **1**([28,29,30]). *Let $K_{n_{1},n_{2},\ldots ,n_{k}}=\left(V_{1},\ldots ,V_{k},E\right)$ be the complete multi-partite graph, where $V_{1},\ldots ,V_{k}$ are disjoint vertex sets, $|V_{i}|=n_{i}$, and 1≤i≤k; each vertex in $V_{i}$ is adjacent to all the vertices in $V\left(K_{n_{1},\ldots ,n_{k}}\right)\backslash V_{i}$. In particular, when $n_{i}=s$, it is called a balanced complete multi-partite graph, and we denote it by K(k,m).*

**Definition** **2**([35,36]). *(The Cartesian product of two graphs) For two graphs*
$G_{1}=\left(V_{1},E_{1}\right)$ and $G_{2}=\left(V_{2},E_{2}\right)$, the Cartesian product graph $G=G_{1}\times G_{2}$ is the graph with vertex set $V_{1}\times V_{2}$. There is an edge from the vertex $\left(x_{1},y_{1}\right)$ to the vertex $\left(x_{2},y_{2}\right)$ if and only if either $x_{1}=x_{2}$ and $y_{1},y_{2}\in E_{2}$ or $y_{1}=y_{2}$ and $x_{1},x_{2}\in E_{1}$.

**Definition** **3**([35]). *(The join of two graphs) The join of simple graphs $G_{1}$ and $G_{2}$, written $G_{1}▿G_{2}$, is the graph obtained from the disjoint union of $G_{1}$ and $G_{2}$ by adding the edges $\left\{xy:x\in V\left(G_{1}\right),y\in V\left(G_{2}\right)\right\}$.*

**Definition** **4**([37,38]). *(The corona of two graphs) Let $G_{1}$ and $G_{2}$ be two graphs on disjoint sets of n and k vertices, respectively. The corona $G_{1}\circ G_{2}$ of $G_{1}$ and $G_{2}$ is defined as the graph obtained by taking one copy of $G_{1}$ and n copies of $G_{2}$, and then joining the ith vertex of $G_{1}$ to every vertex in the ith copy of $G_{2}$, (i=1,2,…,n).*

**Lemma** **1**([34]). *If $G_{1}$ has m vertices and $G_{2}$ has n vertices, then the Laplacian eigenvalues of $G_{1}\times G_{2}$ are the mn numbers: $\nu_{i}\left(G\right)+\nu_{j}\left(H\right)\left(i=1,2...,m;j=1,2,\ldots ,n\right)$, where $\nu_{i}\left(G\right)$ and $\nu_{j}\left(H\right)$ are the Laplacian eigenvalues of $G_{1}$ and $G_{2}$, respectively.*

**Lemma** **2**([34]). *If $G_{1}$ has m vertices and $G_{2}$ has n vertices, then the Laplacian eigenvalues of $G_{1}▿G_{2}$ are: $0,m+n,m+\lambda_{i},n+\mu_{j},i=2,3,\ldots ,n;j=2,3,\ldots ,m$, where $\lambda_{i}$ and $\mu_{j}$ are the non-zero Laplacian eigenvalues of $G_{1}$ and $G_{2}$, respectively.*

### 2.2. Relations for the Coherence and Laplacian Eigenvalues

Refer to references [13,14,15,16,17]. This article considers the first-order noisy network:(1)$$\begin{array}{c} \dot{x}\left(t\right)=-L\left(G\right)x\left(t\right)+\xi \left(t\right), \end{array}$$
where $x\in R^{N}$, and $\xi \left(t\right)\in R^{N}$ is a disturbance vector with zero-mean, unit variance, and a white stochastic noise process.

**Definition** **5**([13,14]). *The first-order network coherence is defined as the mean steady-state variance of the deviation from the average of all node states:*
$$\begin{array}{c} H=\lim_{t\to \infty }\frac{1}{N}\sum_{i=1}^{N}Var\left\{x_{i}\left(t\right)-\frac{1}{N}\sum_{j=1}^{N}x_{j}\left(t\right)\right\}. \end{array}$$

Denote the Laplacian eigenvalues of *G* by $0=\lambda_{1}<\lambda_{2}\leq \ldots \leq \lambda_{N}$. It has been proved that *H* can be characterized by
(2)$$\begin{array}{c} H=\frac{1}{2N}\sum_{i=2}^{N}\frac{1}{\lambda_{i}}. \end{array}$$

The performance index has some similarity with the resistance distance [15,39,40] and Kirhoff index [39,40,41], and other graphical indices [42,43] related to topological structures.

## 3. Main Results

The multilayered networked systems own the graphs composed by connecting the corresponding counterpart nodes of different layers. The following subsections propose the fan-graph-based multilayered structures, starting from the perspective of the multi-partite graph, and then deriving and comparing the first-order coherence of the noisy MASs.

### 3.1. The Coherence for Network Topology $G_{1}\left(a,m,q\right)$

In this subsection, a class of multiplex networks with complete multi-partite structure is considered. The topology can be generated by the operation of complete k-partite graph and fan-graph. Define $G_{1}\left(a,m,q\right):=K\left(a,m\right)\times F_{q}$, and denote the corresponding network with dynamics (1) by $G_{1}$. As shown in Figure 1, from the side view, it has the fan graph structure, and from the horizontal view, each layer is a complete bipartite graph with the partition number a=2, and each partition set has two nodes in it; i.e., m=2. The layer in which each node has the largest vertex degree is named the center layer (see Figure 1), and the vertices in this layer can be viewed as the hub node of the fan graph from the vertical view.

By Lemma 2, one has

$SL\left[K\left(a,m\right)\right]=\left(\begin{array}{ccc} 0 & am & (a-1)m\\ 1 & a-1 & am-a \end{array}\right)$, and since

$SL\left(F_{q}\right)=\left(\begin{array}{ccc} 0 & q & 1+4sin^{2}\left(\frac{k\pi }{2(q-1)}\right)\\ 1 & 1 & q-2 \end{array}\right)$.

By Lemma 1, it can be derived that $SL[G_{1}\left(a,m,q\right)]$ has the description:(1).$0\in SL[G_{1}\left(a,m,q\right)]$ with multiplicity 1;(2).$am\in SL(G_{1}\left(a,m,q\right))$ with multiplicity (a−1);(3).$\left(a-1\right)m\in SL(G_{1}\left(a,m,q\right))$ with multiplicity a(m−1);(4).$q\in SL(G_{1}\left(a,m,q\right))$ with multiplicity 1;(5).$q+am\in SL(G_{1}\left(a,m,q\right))$ with multiplicity a−1;(6).$q+\left(a-1\right)m\in SL(G_{1}\left(a,m,q\right))$ with multiplicity a(m−1);(7).$1+4sin^{2}\frac{k\pi }{2(q-1)}\in SL\left(G_{1}\left(a,m,q\right)\right)$ with multiplicity 1, where k=1,2,…,q−2;(8).$am+1+4sin^{2}\frac{k\pi }{2(q-1)}\in SL\left(G_{1}\left(a,m,q\right)\right)$ with multiplicity a−1, k=1,2,…,q−2;(9).$\left(a-1\right)m+1+4sin^{2}\frac{k\pi }{2(q-1)}\in SL\left(G_{1}\left(a,m,q\right)\right)$ with multiplicity a(m−1), k=1,2,…,q−2.

Therefore, we have
$$\begin{array}{cc} H^{\left(1\right)}\left(G_{1}\right)= & \frac{1}{2amq}(\left(a-1\right)\frac{1}{am}+\left(am-a\right)\frac{1}{(a-1)m}+\frac{1}{q}+\left(a-1\right)\frac{1}{q+am}\\ & +\left(am-a\right)\frac{1}{q+(a-1)m}+\sum_{k=1}^{q-2}\frac{1}{1+4sin^{2}\frac{k\pi }{2(q-1)}}\\ & +\left(a-1\right)\sum_{k=1}^{q-2}\frac{1}{am+1+4sin^{2}\frac{k\pi }{2(q-1)}}+\left(am-a\right)\sum_{k=1}^{q-2}\frac{1}{\left(a-1\right)m+1+4sin^{2}\frac{k\pi }{2(q-1)}}). \end{array}$$

(i). If a,m are fixed, let q→∞, then one can get that:$$\begin{array}{c} \lim_{q\to \infty }H\left(G_{1}\right)=\frac{\sqrt{5}}{10am}+\frac{\left(a-1\right)}{2am\sqrt{\left(am+5)(am+1\right)}}+\frac{\left(am-a\right)}{2am\sqrt{\left[(a-1)m+5][(a-1)m+1\right]}}, \end{array}$$
and hence, by contraction of inequality, when *a* is large enough, $H(G_{1})\to 0$; when *m* is large enough, $H(G_{1})\to 0$.

(ii). If a,q are fixed, let m→∞. Then, $H(G_{1})\to 0$.

(iii). Suppose that m,q are fixed, let a→∞, and then $H(G_{1})\to 0$ also holds.

### 3.2. The Coherence for Network Topology $G_{2}\left(n,p,q\right)$

As shown in Figure 2, from the horizontal vision, each layer is composed by the join of the complete graph and the empty graph. The complete subgraph can be viewed as a generation of the concept of the hub node of a star graph. From the side view, it can be seen as the fan graph structures. Mathematically, define $G_{2}\left(n,p,q\right):=\left(K_{n}▿E_{p}\right)\times F_{q}$, and the corresponding noisy network is denoted by $G_{2}$. It can be derived that the Laplacian spectrum has the following characterization:(1).$0\in SL[G_{2}\left(n,p,q\right)]$ with multiplicity 1;(2).$n+p\in SL[G_{2}\left(n,p,q\right)]$ repeated *n* times;(3).$n\in SL[G_{2}\left(n,p,q\right)]$ repeated p−1 times;(4).$q\in SL[G_{2}\left(n,p,q\right)]$ with multiplicity 1;(5).$n+p+q\in SL[G_{2}\left(n,p,q\right)]$ with multiplicity *n*;(6).$n+q\in SL[G_{2}\left(n,p,q\right)]$ repeated p−1 times;(7).$1+4sin^{2}\left(\frac{k\pi }{2(q-1)}\right)\in SL\left[G_{2}\left(n,p,q\right)\right]$ with multiplicity 1, k=1,2,…,q−2;(8).$n+p+1+4sin^{2}\left(\frac{k\pi }{2(q-1)}\right)\in SL\left[G_{2}\left(n,p,q\right)\right]$ with multiplicity *n*, k=1,2,…,q−2;(9).$n+1+4sin^{2}\left(\frac{k\pi }{2(q-1)}\right)\in SL\left[G_{2}\left(n,p,q\right)\right]$ repeated p−1 times, k=1,2,…,q−2.

Therefore,
$$\begin{array}{cc} H(G_{2})= & \frac{1}{2(n+p)q}[\frac{n}{n+p}+\frac{p-1}{n}+\frac{1}{q}+\frac{n}{n+p+q}+\frac{p-1}{n+q}+\sum_{k=1}^{q-2}\frac{1}{1+4sin^{2}\left(\frac{k\pi }{2(q-1)}\right)}\\ & +n\sum_{k=1}^{q-2}\frac{1}{n+p+1+4sin^{2}\left(\frac{k\pi }{2(q-1)}\right)}+\left(p-1\right)\sum_{k=1}^{q-2}\frac{1}{n+1+4sin^{2}\left(\frac{k\pi }{2(q-1)}\right)}]. \end{array}$$

Thus, we have,

(i).
$$\begin{array}{cc} \lim_{q\to \infty }H\left(G_{2}\right)= & \frac{1}{2(n+p)}\int_{0}^{1}\frac{1}{1+4sin^{2}\left(\frac{\pi x}{2}\right)}dx+\frac{n}{2(n+p)}\int_{0}^{1}\frac{1}{n+p+1+4sin^{2}\left(\frac{\pi x}{2}\right)}dx\\ & +\frac{p-1}{2(n+p)}\int_{0}^{1}\frac{1}{n+1+4sin^{2}\left(\frac{\pi x}{2}\right)}dx\\ = & \frac{\sqrt{5}}{10(n+p)}+\frac{n}{2(n+p)}\frac{1}{\sqrt{\left(n+p+5)(n+p+1\right)}}+\frac{p-1}{2(n+p)}\frac{1}{\sqrt{\left(n+5)(n+1\right)}} \end{array}$$

(ii). When p→∞, $H\left(G_{2}\right)\to \frac{1}{2nq}+\frac{1}{2(n+q)q}+\frac{1}{2q}\sum_{k=1}^{q-2}\frac{1}{n+1+4sin^{2}\frac{k\pi }{2(q-1)}}$;

(iii). When n→∞, $H(G_{2})\to 0$.

**Remark** **1.**
*If the numbers of vertices of each layer of $G_{1}$ and $G_{2}$ are equal, that is, am=n+p, then the sufficient condition that makes $\lim_{q\to \infty }H\left(G_{1}\right)\geq \lim_{q\to \infty }H\left(G_{2}\right)$ hold is acquired: $\frac{n+p-a}{n+p-m+5}\geq \frac{p-1}{n+1}$, where a,m,n,p≥2, and the parameters are all integer.*


### 3.3. The Coherence of Structure $G_{3}\left(n,q,l\right)$

In this case, a class of multi-layered star-composed structure is considered. Each layer can be viewed as several star-shaped copies formed into the complete subgraph by the center vertices (see Figure 3), and all layers are connected only through black counterpart nodes. It can be seen as the fan graph from the side view perspective. Define $G_{3}\left(n,q,l\right)=\left(K_{n}\times F_{q}\right)\circ E_{l}$, and denote the corresponding noisy network by $G_{3}$.

Since
$$SL\left(K_{n}\times F_{q}\right)=\left(\begin{array}{cccccc} 0 & q & 1+4sin^{2}\frac{k\pi }{2(q-1)} & n & n+q & n+1+4sin^{2}\frac{k\pi }{2(q-1)}\\ 1 & 1 & 1 & n-1 & n-1 & n-1 \end{array}\right),$$
where k=1,2,…,q−2.

$SL[G_{3}\left(n,q,l\right)]$ has the following characterization:(1).0 and $l+1\in SL[G_{3}\left(n,q,l\right)]$ repeated once;(2).$\frac{q+l+1\pm \sqrt{{\left(q+l+1\right)}^{2}-4q}}{2}\in SL\left[G_{3}\left(n,q,l\right)\right]$ with multiplicity 1;(3).$\frac{\left(1+4sin^{2}\frac{k\pi }{2(q-1)}+l+1\right)\pm \sqrt{{\left(1+4sin^{2}\frac{k\pi }{2(q-1)}+l+1\right)}^{2}-4\left(1+4sin^{2}\frac{k\pi }{2(q-1)}\right)}}{2}\in SL\left[G_{3}\left(n,q,l\right)\right]$ repeated once, where k=1,2,…,q−2.(4).$\frac{n+l+1\pm \sqrt{{\left(n+l+1\right)}^{2}-4n}}{2}\in SL\left[G_{3}\left(n,q,l\right)\right]$ repeated n−1 times;(5).$\frac{n+q+l+1\pm \sqrt{{\left(n+q+l+1\right)}^{2}-4\left(n+q\right)}}{2}\in SL\left[G_{3}\left(n,q,l\right)\right]$ repeated n−1 times;(6).$\frac{\left(n+1+4sin^{2}\frac{k\pi }{2(q-1)}+l+1\right)\pm \sqrt{{\left(n+1+4sin^{2}\frac{k\pi }{2(q-1)}+l+1\right)}^{2}-4\left(n+1+4sin^{2}\frac{k\pi }{2(q-1)}\right)}}{2}\in SL\left[G_{3}\left(n,q,l\right)\right]$ repeated n−1 times, where k=1,2,…,q−2.(7).$1\in SL[G_{3}\left(n,q,l\right)]$ repeated qn(l−1) times.

Therefore,
$$\begin{array}{cc} H(G_{3})= & \frac{1}{2nq(1+l)}(\frac{1}{l+1}+\frac{q+l+1}{q}+\sum_{k=1}^{q-2}\left(1+\frac{l+1}{1+4sin^{2}\frac{k\pi }{2(q-1)}}\right)+\frac{\left(n+l+1)(n-1\right)}{n}\\ & +\frac{\left(n+q+l+1)(n-1\right)}{n+q}+\sum_{k=1}^{q-2}\frac{\left(n+1+4sin^{2}\frac{k\pi }{2(q-1)}+l+1\right)\left(n-1\right)}{n+1+4sin^{2}\frac{k\pi }{2(q-1)}}+qn\left(l-1\right)), \end{array}$$
and then we have

(i). When q→∞, $H\left(G_{3}\right)\to \frac{1}{2n}\int_{0}^{1}\frac{1}{1+4sin^{2}\frac{\pi x}{2}}dx+\frac{n-1}{2n}\int_{0}^{1}\frac{1}{n+1+4sin^{2}\frac{\pi x}{2}}dx+\frac{l}{2(1+l)}=\frac{\sqrt{5}}{10n}+\frac{n-1}{2n}\frac{1}{\sqrt{\left(n+5)(n+1\right)}}+\frac{l}{2(1+l)}$;

(ii). When n→∞, $H\left(G_{3}\right)\to \frac{l}{2(1+l)}$;

(iii). When l→∞,

$H\left(G_{3}\right)\to \frac{1}{2nq^{2}}+\frac{1}{2nq}\sum_{k=1}^{q-2}\frac{1}{1+4sin^{2}\frac{k\pi }{2(q-1)}}+\frac{n-1}{2n^{2}q}+\frac{n-1}{2nq(n+q)}+\frac{1}{2nq}\sum_{k=1}^{q-2}\frac{n-1}{n+1+4sin^{2}\frac{k\pi }{2(q-1)}}$.

**Remark** **2.**
*Both $G_{2}$ and $G_{3}$ have the complete substructure $K_{n}$, and the number of layers in both cases is q. If the node numbers of one layer in $G_{2}$ and $G_{3}$ are equal, i.e., n+nl=n+p, that is, nl=p, it can be derived that $\lim_{q\to \infty }H\left(G_{3}\right)>\lim_{q\to \infty }H\left(G_{2}\right)$ holds.*


### 3.4. The Coherence for Special Cases

The Laplacian spectrum of star graph with *p* vertices is:

$SL\left(S_{p}\right)=\left(\begin{array}{ccc} 0 & p & 1\\ 1 & 1 & p-2 \end{array}\right)$,

Def. $W_{1}=S_{p}\times F_{n}$, and the related noisy network is denoted by $W_{1}$; then we have $SL(W_{1})$
$$=\left(\begin{array}{ccccccccc} 0 & n & 1+4sin^{2}\left(\frac{k\pi }{2(n-1)}\right) & p & p+n & p+1+4sin^{2}\left(\frac{k\pi }{2(n-1)}\right) & 1 & 1+n & 2+4sin^{2}\left(\frac{k\pi }{2(n-1)}\right)\\ 1 & 1 & 1 & 1 & 1 & 1 & p-2 & p-2 & p-2 \end{array}\right)$$
where k=1,2,…,n−2.

The first-order coherence for network $W_{1}$:$$\begin{array}{cc} H(W_{1})= & \frac{1}{2np}(\frac{1}{n}+\sum_{k=1}^{n-2}\frac{1}{1+4sin^{2}\left(\frac{k\pi }{2(n-1)}\right)}+\frac{1}{p}+\frac{1}{p+n}+\sum_{k=1}^{n-2}\frac{1}{p+1+4sin^{2}\left(\frac{k\pi }{2(n-1)}\right)}\\ & +\left(p-2\right)+\frac{p-2}{1+n}+\left(p-2\right)\sum_{k=1}^{n-2}\frac{1}{2+4sin^{2}\left(\frac{k\pi }{2(n-1)}\right)}), \end{array}$$
thus,
$$\begin{array}{cc} \lim_{n\to \infty }H^{\left(1\right)}= & \frac{1}{2p}\int_{0}^{1}\frac{1}{1+4sin^{2}\left(\frac{\pi x}{2}\right)}dx+\frac{1}{2p}\int_{0}^{1}\frac{1}{p+1+4sin^{2}\left(\frac{\pi x}{2}\right)}dx+\frac{p-2}{2p}\cdot \frac{\sqrt{3}}{6}\\ = & \frac{\sqrt{5}}{10p}+\frac{1}{2p}\frac{1}{\sqrt{\left(p+5)(p+1\right)}}+\frac{p-2}{2p}\cdot \frac{\sqrt{3}}{6} \end{array}$$
Define a noisy network $W_{2}$ with the graph $W_{2}:=K_{p}\times F_{n}$; then we have
$$\begin{array}{c} SL\left(W_{2}\right)=\left(\begin{array}{cccccc} 0 & n & 1+4sin^{2}\left(\frac{k\pi }{2(n-1)}\right) & p & p+n & p+1+4sin^{2}\left(\frac{k\pi }{2(n-1)}\right)\\ 1 & 1 & 1 & p-1 & p-1 & p-1 \end{array}\right) \end{array}$$
where k=1,2,…,n−2.

Therefore,
$$\begin{array}{cc} H(W_{2})= & \frac{1}{2np}(\frac{1}{n}+\sum_{k=1}^{n-2}\frac{1}{1+4sin^{2}\left(\frac{k\pi }{2(n-1)}\right)}+\frac{p-1}{p}+\frac{p-1}{p+n}\\ & +\left(p-1\right)\sum_{k=1}^{n-2}\frac{1}{p+1+4sin^{2}\left(\frac{k\pi }{2(n-1)}\right)}). \end{array}$$

Then one has
$$\begin{array}{cc} \lim_{n\to \infty }H\left(W_{2}\right)= & \frac{1}{2p}\int_{0}^{1}\frac{1}{1+4sin^{2}\left(\frac{\pi x}{2}\right)}dx+\frac{p-1}{2p\sqrt{\left(p+5)(p+1\right)}}\\ = & \frac{\sqrt{5}}{10p}+\frac{p-1}{2p\sqrt{\left(p+5)(p+1\right)}} \end{array}$$

**Remark** **3.**
*From the above derivation, it can be acquired that $H\left(W_{1}\right)\to \frac{\sqrt{3}}{12}$, as p is large enough, and $H(W_{2})\to 1$ as p is large enough. For the comparison of the three networks, suppose that am=p; i.e., the vertex sets of counterpart layers have the same cardinality. In real networks that may have the multi-partite layered structures, one reasonable hypothesis is that p→+∞, usually corresponding to the trend that one of the parameters a and m tends to infinity.*


**Remark** **4.**
*The implication of this research is that the mathematical expression of coherence can be a reference for deriving the robustness of similar MASs with multi-layered structures. The asymptotic results has practical significance to improving the robustness of the related network. In addition, the fan-graph from the vertical view, which forms a center layer structurally, might enlighten the future research on the topological design of multi-layer networks.*


## 4. Simulation and Comparison

This section presents the simulation and comparisons of the coherence for these multi-layered MASs. Figure 4 shows the changes in the coherence with the parameters *a* and *q*. These results coincide with the results that Section 3.1 shows; i.e., if *m* is fixed, then $H(G_{1})\to 0$ as q,a→∞. It coincides with results (i) and (iii) in Section 3.1. In Figure 5, the surface is for $H(G_{1})$ with the changes in *m* and *q*, when a=3. The figure implies that $H(G_{1})\to 0$ as m→∞, which satisfies results (i) and (ii) in Section 3.1. Figure 6 shows the changes in coherence of $G_{1}$ and $G_{2}$ with the changing of parameter *q*, i.e., the number of layers; the node related parameters were chosen as: a=3,m=8,n=18, and p=6. These satisfy the condition that Remark 2 mentioned: $\frac{n+p-a}{n+p-m+5}\geq \frac{p-1}{n+1}$. One can see that $H(G_{1})$ is a bit larger than $H(G_{2})$ in the range of all *q*, which satisfies the previous result. Figure 7 shows the changes in $H(G_{2})$ and $H(G_{3})$ with *q*, when the other parameters were chosen as n=3,l=3,p=nl=9. Figure 8 describes the variation in $H(G_{1})$ with *m*, when a=3,q=4. From the point (197, 0.001469) and the decreasing trend of the curve, one can see that when m→∞, the coherence $H(G_{1})\to 0$ coincides with the result in Section 3.1. In Figure 9, the point (194, 0.001072) and the curve reflect the change trends of $H(G_{1})$ with *a*, when m=3,q=4; that is, $H(G_{1})\to 0$. This also coincides with the previous result. Figure 10 describes the monotonous increasing trend of $H(G_{2})$ with *p*. From the former section, we know that when *n* and *q* are fixed, $H(G_{2})$ will tend to a certain value, and the point (196, 0.101) satisfies the result. This coincides with the figure’s description. In Figure 11, the curve and the point (191, 0.003161) shows the trend that $H(G_{2})$ monotonously decreases to 0 when n→∞. Figure 12 implies the decreasing trend that $H(G_{3})$ will tend to a certain value when n→∞, and the certain fixed value is related only to *l*. In this figure, q=4,l=3; the point (193,0.3781) coincides with the result that the former section derived. In Figure 13, the increasing curve implies the trend that $H(G_{3})$ will tend to a certain value which is related to *n* and *q*; at l→∞, the point (196, 0.6072) is consistent with this fact. One can see that the points on the curves satisfy both the mathematical expression of coherence and the asymptotic results. The simulations also verified the size relation of the three cases when the corresponding layers have the same number of nodes. In summary, the mathematical expression results, the integral calculations, and the simulation with partially fixed parameters are consistent.

## 5. Conclusions

In this research, the first-order coherence for several multi-layered MASs with complete, multi-partite graph structures was studied. A fan graph was used to define the novel multi-layered structures. The view from the side perspective shows that each has the fan-graph structure. The approaches of analysis and graph spectra were combined to derive the mathematical expression of a performance index, and the corresponding asymptotic properties have been acquired. We find that when the cardinality of a node set of corresponding counterpart network layers is consistent, the multi-layered topology class with complete multi-partite structure has the best robustness of all the considered systems, if the sufficient conditions for the parameters hold. Finally, the coherence of the non-isomorphic multi-layered networks with the same number of nodes in their counterpart layers were simulated to verify the results.

## Figures and Tables

**Figure 1 entropy-25-00040-f001:**
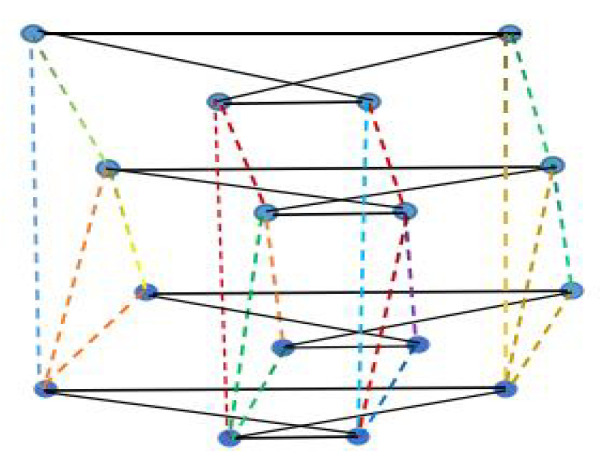
An example of $G_{1}\left(a,m,q\right)$ with a=2,m=2,q=4.

**Figure 2 entropy-25-00040-f002:**
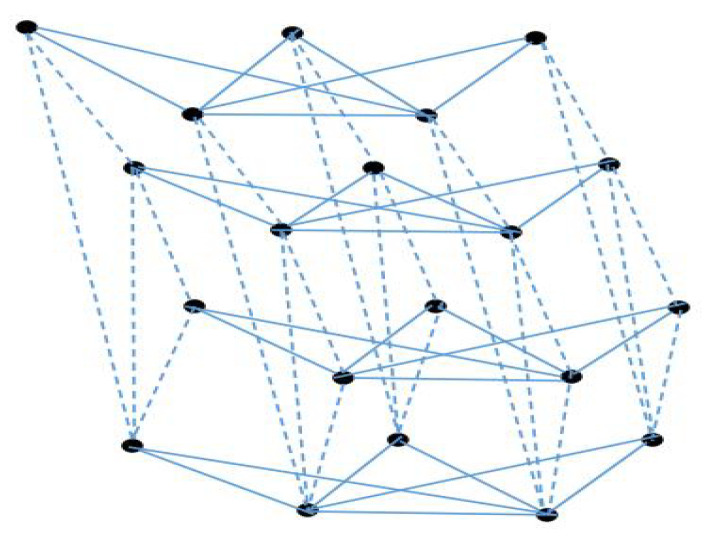
An example of $G_{2}\left(n,p,q\right)$ with n=2,p=3,q=4.

**Figure 3 entropy-25-00040-f003:**
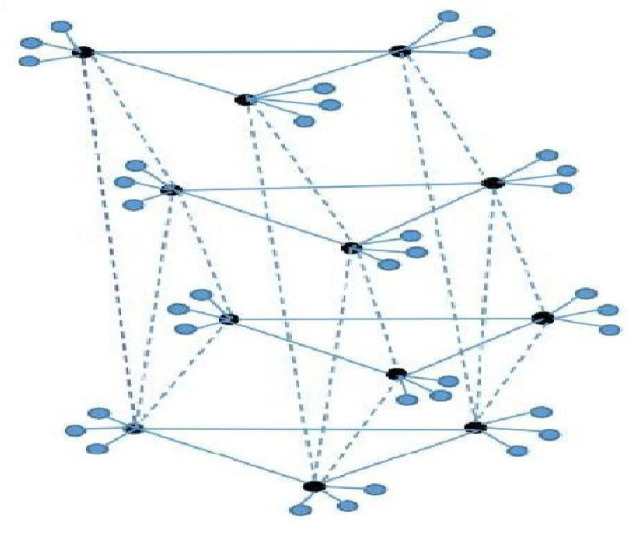
An example of $G_{3}\left(n,q,l\right)$ with n=3,q=4,l=3.

**Figure 4 entropy-25-00040-f004:**
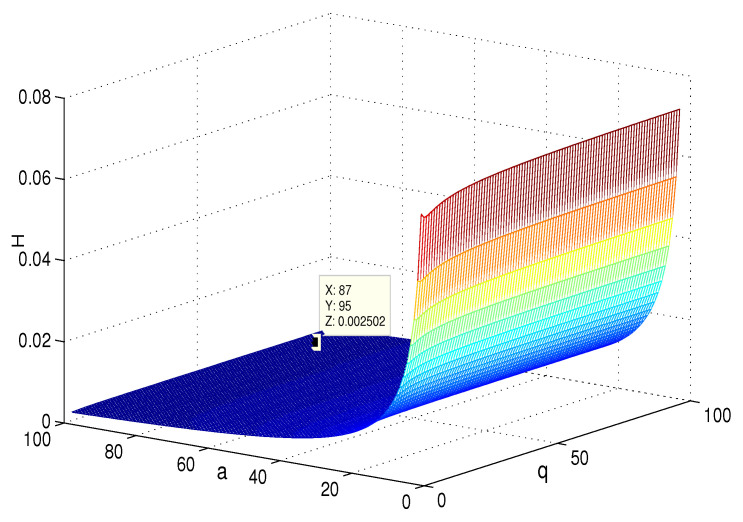
The change in $H(G_{1})$ with *a* and *q*; m=3.

**Figure 5 entropy-25-00040-f005:**
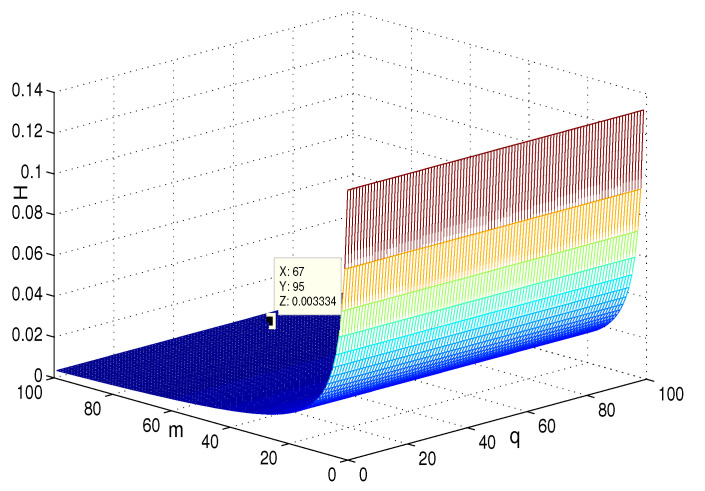
The change in $H(G_{1})$ with *m* and *q*; a=3.

**Figure 6 entropy-25-00040-f006:**
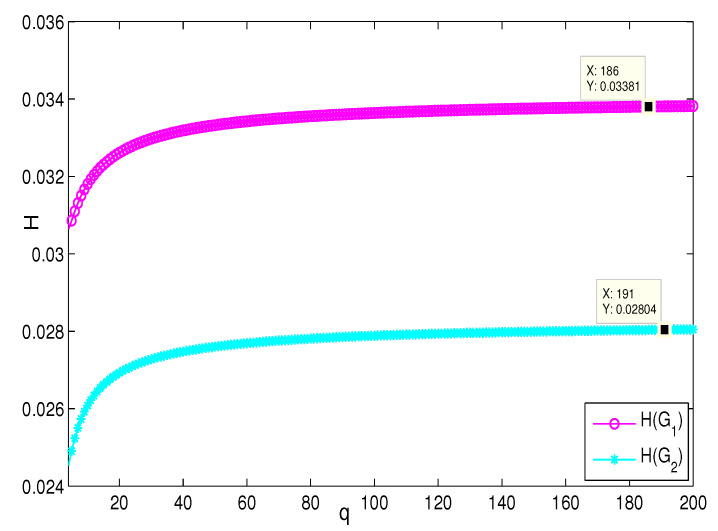
The change in $H(G_{1})$ and $H(G_{2})$ with *q*; a=3,m=8,n=18,p=6.

**Figure 7 entropy-25-00040-f007:**
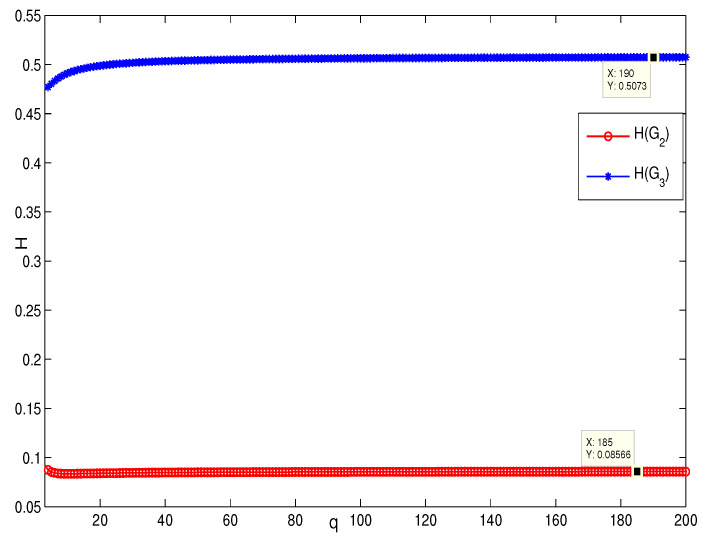
The change in $H(G_{2})$ and $H(G_{3})$ with *q*; n=3,l=3,p=9.

**Figure 8 entropy-25-00040-f008:**
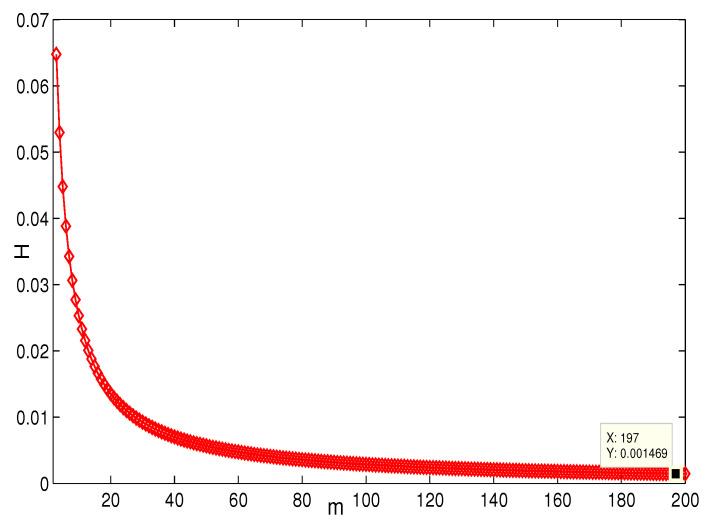
The change in $H(G_{1})$ with *m*. a=3,q=4.

**Figure 9 entropy-25-00040-f009:**
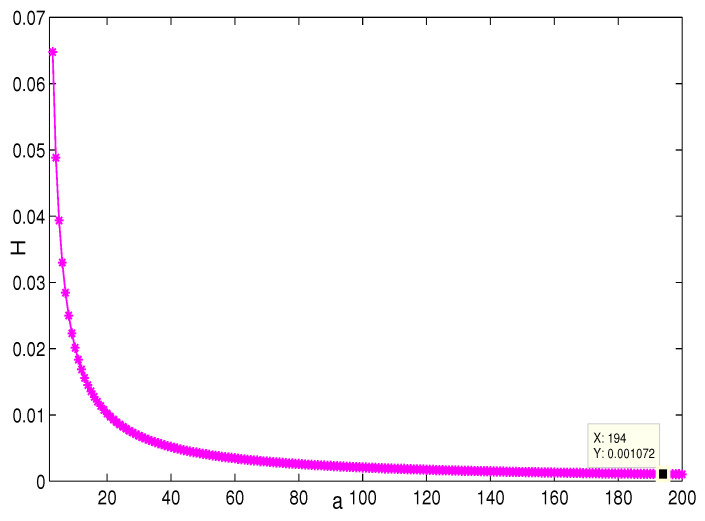
The change in $H(G_{1})$ with *a*. m=3,q=4.

**Figure 10 entropy-25-00040-f010:**
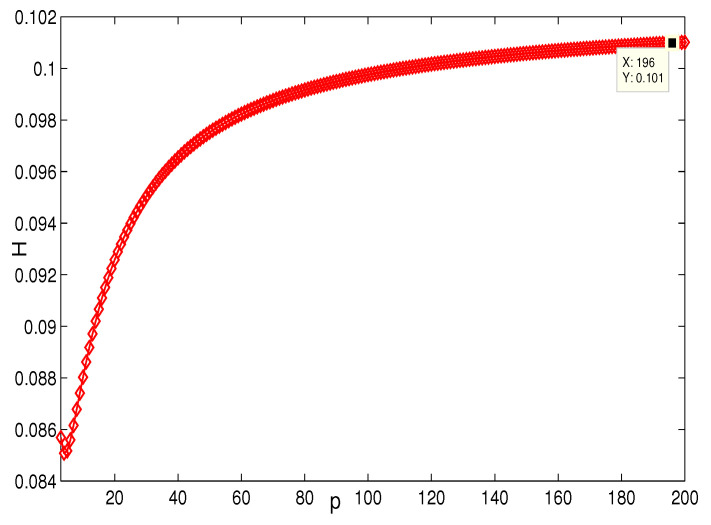
The change in $H(G_{2})$ with *p*. n=3,q=4.

**Figure 11 entropy-25-00040-f011:**
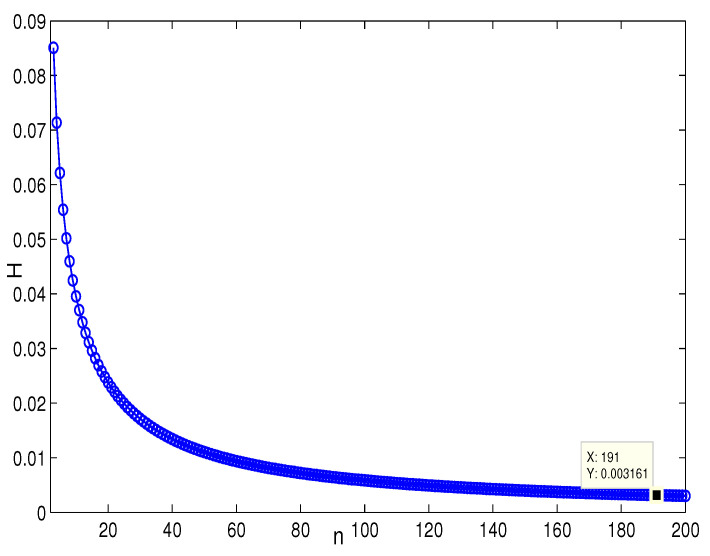
The change in $H(G_{2})$ with *n*. p=4,q=4.

**Figure 12 entropy-25-00040-f012:**
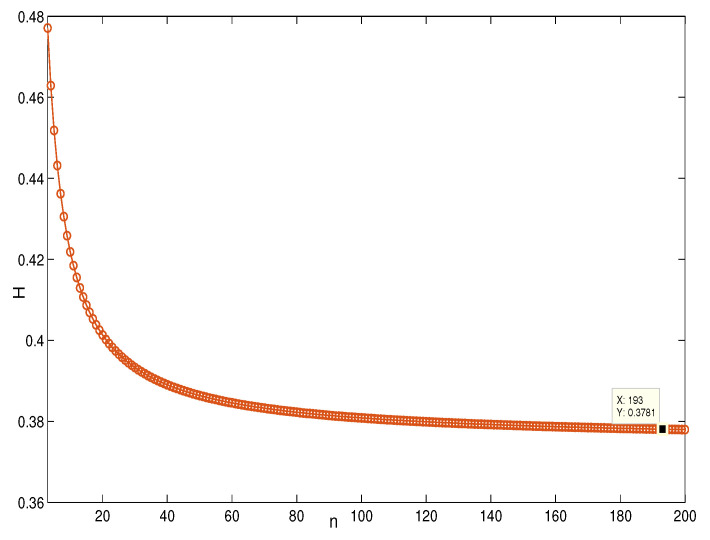
The change in $H(G_{3})$ with *n*. q=4,l=3.

**Figure 13 entropy-25-00040-f013:**
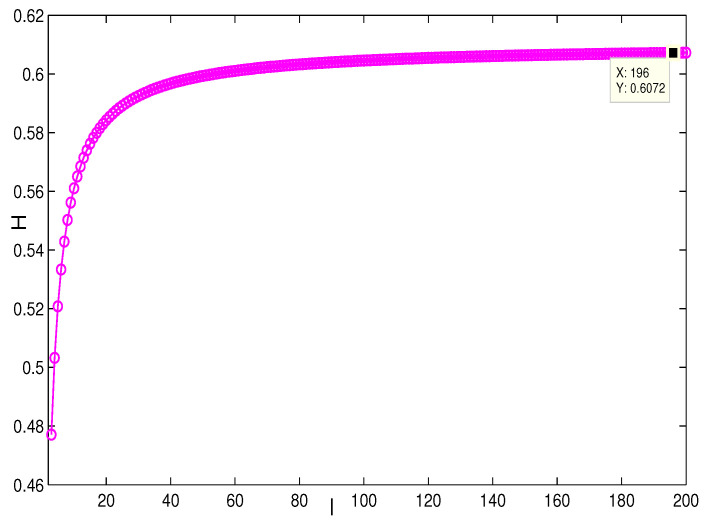
The change in $H(G_{3})$ with *l*. n=3,q=4.

## Data Availability

Not applicable.

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
