# Peer review of "On the Consensus Performance of Multi-Layered MASs with Various Graph Parameters—From the Perspective of Cardinalities of Vertex Sets"

_entropy, 2022, doi:10.3390/e25010040_

Round 1
Reviewer 1 Report
The authors use established formulas and techniques to obtain exact and asymptotic expressions for the coherence, i.e. the variance of the components for a multivariate Ornstein-Uhlenbeck process in several classes of multi-layered networks. The coherence is expressed as the average of the inverses of the non-zero Laplacian eigenvalues. Thus presenting the Laplacian spectrum for some classes of networks, the coherence can be calculated and the limits explored. The authors seem to use the term performance index synonymously with coherence but do not say so explicitly. Actually zero coherence H would mean the best coherence and performance, no?
The authors use symmetry to derive the Laplacian spectrum for various product graphs from the spectra of the component graphs. I was not familiar with the join operation and the corona of two graphs, nor do I know how the spectrum of the product graph is derived from the spectra of the component graphs. A separate section on this technique with appropriate citations would be very helpful.
In Sec.3.1 the components spectra of G1 are written in table/matrix form but the spectrum of G1 is described by text in a whole paragraph with inline formulas. This makes it harder to oversee the result. Same for Sec.3.2 and graph class G2. In Sec.3.3 there is at least a numbered list, but a table would maybe be more clear. Please give citations for the component spectra or derive them. The authors write "By the corresponding characteristic polynomial, one has ....". One can believe that or not.
Since I am unfamiliar with some of the joining operations, the pictures Fig.1-3 of the graphs from the three classes G1,G2 and G3 are somewhat helpful. But the relation of the class parameters and the picture is not so clear. It would also be helpful to see the components graphs in the same picture, or somehow better mark the substructures in the example graphs. Especially the general structure of G2 is not clear.
The three graph classes and the special cases W1 and W2 (with pictures please) are somewhat arbitrary. That is why a section on the derivation of the product graph spectra would be so much more helpful and more general than the results for just these classes.
Is the comparison with simulations really necessary? Figures 4-13 are maybe a bit repetitive and do not carry much information. In Figs.6 and 7 coherence of different graph classes with the same fan graph motives (q) are compared. I am not sure if such a comparison makes much sense, nor is it well discussed in the text. Asymptotics should be marked by horizontal lines (from the formulas) and not by a data point near the end. What is happening in Fig.10 for small q? Is that a simulation artifact or is there actually an anti-coherence-resonance?
I would recommend the publication of these results, which I find interesting, after the points above have been addressed. However, the English language in this manuscript is really subpar. It reads like an automatic translation with some awkward expressions. I strongly recommend to use a proof reading service.
Author Response
Thanks very much for your comments, please see the attachments.

Reviewer 2 Report
Please find the attachment: Report_entropy-2005705.

Author Response
Thanks very much for your comments, please see the attachments

Reviewer 3 Report
The authors study well-known spectral methods to derive mathematical expressions for the coherence index of these graphs. I find the aim of the paper particularly challenging, but after that, the authors do not discuss the implications of the work nor present considerable results to support their claims.
The introduction of the manuscript must be seriously reworked to contain at least the main references of network theory of the last 20 years. Many overlooked works are not included in the introduction and discussion, while other minor works are profoundly discussed.
In my opinion, the paper can be considered for publication only after a major revision that clarifies the implications, reworking results and figures, and the fit on the network state-of-the art of their work.
Author Response

(The authors gave the same response as above.)

Round 2
Reviewer 1 Report
- please include the definitions of the joining operations (join, cartesian and corona) from your response in the manuscript.
- please include all relevant lemma for the Laplacian spectra of composite graphs (join, cartesian and corona) in the manuscript, with appropriate citation. (Laplacian spectrum for a symmetric 0-1 adjacency matrix)
- in lemma 2 and 3 of the authors response to my comments (lemma 1 in the manuscript) the Laplacian eigenvalues need to be introduced before they are used in the formula. In Lemma 2 the eigenvalues of the second graph should also have a different symbol (e.g. mu instead of nu).
- please define the noise strength in Eq.1. Different definitions of noise strength may vary by a factor of two.
- the language of the manuscript is still not acceptable for publication. MDPI offers an editing service for English grammar and expression https://www.mdpi.com/authors/english.
Author Response
Dear professor,
Thanks very much for your good comments, the article has been improved a lot
thanks to your valuable comments and suggestion. Please see the attachments.

Round 3
Reviewer 1 Report
I am glad that the authors have carefully followed my suggestions, including using a professional editing service. The quality of the manuscript has improved and is now adequate for publication.